# Fabrication and Characterization of Chitosan–Pea Protein Isolate Nanoparticles

**DOI:** 10.3390/molecules27206913

**Published:** 2022-10-14

**Authors:** Man Zhang, Zikun Li, Mengqi Dai, Hongjun He, Bin Liang, Chanchan Sun, Xiulian Li, Changjian Ji

**Affiliations:** 1College of Life Sciences, Yantai University, Yantai 264005, China; 2College of Food Engineering, Ludong University, Yantai 264025, China; 3School of Pharmacy, Binzhou Medical University, Yantai 264003, China; 4Department of Physics and Electronic Engineering, Qilu Normal University, Jinan 250200, China

**Keywords:** pea protein isolate, chitosan, nanoparticles, interface properties

## Abstract

Chitosan (CS) and pea protein isolate (PPI) were used as raw materials to prepare nanoparticles. The structures and functional properties of the nanoparticles with three ratios (1:1, 1:2 1:3, CS:PPI) were evaluated. The particle sizes of chitosan–pea protein isolate (CS–PPI) nanoparticles with the ratios of 1:1, 1:2, and 1:3 were 802.95 ± 71.94, 807.10 ± 86.22, and 767.75 ± 110.10 nm, respectively, and there were no significant differences. Through the analysis of turbidity, endogenous fluorescence spectroscopy and Fourier transform infrared spectroscopy, the interaction between CS and PPI was mainly caused by electrostatic mutual attraction and hydrogen bonding. In terms of interface properties, the contact angles of nanoparticles with the ratio of 1:1, 1:2, and 1:3 were 119.2°, 112.3°, and 107.0°, respectively. The emulsifying activity (EAI) of the nanoparticles was related to the proportion of protein. The nanoparticle with the ratio of 1:1 had the highest potential and the best thermal stability. From the observation of their morphology by transmission electron microscopy, it could be seen that the nanoparticles with a ratio of 1:3 were the closest to spherical. This study provides a theoretical basis for the design of CS–PPI nanoparticles and their applications in promoting emulsion stabilization and the delivery of active substances using emulsions.

## 1. Introduction

Chitosan (CS) is the only cationic polysaccharide in nature and exists in the deacetylated form of chitin. It has great biodegradability, biocompatibility, and is environmentally friendly [1]. Due to its unique charge, CS can bind to negatively charged proteins through electrostatic interactions in a specific pH range [2]. In addition, CS participating in the encapsulation of active substances can resist the destruction of the strong acidity of gastric juice, which protects the active substances and improves their bioavailability [3].

Pea protein isolate (PPI) is a natural protein, which has certain emulsifying properties, foaming properties and water-holding properties. Further, compared with soy protein, it has lower allergenicity and is safer [4]. PPI has gradually become a substitute for soy protein. It is generally extracted by the alkali-soluble acid precipitation method [3]. Its isoelectric point is about 4.3. The solubility of PPI is the worst when the pH is near the isoelectric point and is better under alkaline conditions. The composition of PPI includes globulin, albumin, gliadin and gluten [5], of which globulin accounts for the largest proportion accounting for about 65–80%. Globulin is composed of legumin (11 S) and vicilin (7 S) [6]. In addition, PPI is more sensitive near the isoelectric point and prone to aggregation [7]. Studies have shown that PPI can be modified by physical, chemical and enzymatic modification methods to improve various functional properties. Sun et al. (2015) found that ultrafine pulverization and micronization of whey protein isolate improved its hydrophobicity and enabled it to have excellent emulsion stability [8].

There are two types of binding between proteins and polysaccharides: covalent bonds and noncovalent bonds. Noncovalent bonds mainly include electrostatic interactions, hydrogen bonds, and hydrophobic interactions [7,9]. Covalent bonding is mainly through the Maillard reaction to form carbonyl amino bonds. Studies have shown that the binding of proteins and polysaccharides can be affected by changes in external conditions. Okagu et al. (2021) found that the use of succinylation to modify protein can increase the negative charge on the protein surface, which is conducive to the tight binding of the protein to positively charged polysaccharides [3]. In addition, calcium ions can form Ca^2+^ bridges through the electrostatic interaction between PPI and HMP molecules. The dense structure formed by Ca^2+^ bridges enhances the stability of HMP molecular chains, and the addition of calcium ions promotes the formation of protein and polysaccharide complexes [7]. The complex formed by the combination of protein and polysaccharides can improve the deficiency of protein alone. Yuan et al. (2013) found that the formation of glycinin/CS soluble complexes at an acidic pH could improve the interfacial and emulsifying properties of glycinin [10]. Nanoemulsions prepared from Maillard conjugates formed from casein hydrolyzates and carboxymethyl chitosan have higher stability than pure casein. The conjugated polysaccharides provide strong steric effects. The freeze-thaw and pH stability of the prepared O/W nanoemulsions can be greatly improved by hydrolysis and polysaccharide grafting modification [11]. The combination of low methoxyl pectin with whey protein isolate (WPI) can enhance the thermal stability of oil-in-water emulsions stabilized only by whey protein [12]. In addition to improving the stability of Pickering emulsions, protein and polysaccharide nanocomplexes and Pickering emulsions composed of them also have good encapsulation capabilities. Some active ingredients, such as curcumin, resveratrol, etc., are mostly low in bioavailability due to poor water solubility or being easily affected by the external environment. Compared with individual components, protein and polysaccharide complexes have higher encapsulation abilities and protective effects on active components [7,13]. The oil-in-water emulsion prepared from whey protein isolate and the chitosan complex can well encapsulate α-tocopherol and the encapsulation rate can reach up to 86.3%; a controlled and sustainable release can be achieved in the digestive system [14].

In this study, CS and PPI were selected as raw materials, and the effect of volume ratio on the formation and functional properties of CS–PPI nanoparticles were mainly explored. The particle size, zeta potential and turbidity of CS–PPI nanoparticles with three different ratios (1:1, 1:2, 1:3) were measured and compared. Through the analysis of the endogenous fluorescence spectrum and Fourier transform infrared spectrum, the interaction and the interaction force between CS and PPI were obtained and analyzed. The interfacial properties of nanoparticles can be analyzed through the study of contact angle and emulsification. The thermal stability of nanoparticles was evaluated by comparing the change in particle size before and after heating. Furthermore, the microscopic morphology of the particles was visually observed by transmission electron microscopy. Through the research and analysis obtained from this study, it will be helpful to deeply understand the binding mode between CS and PPI and the effect of protein concentration on the formation of CS–PPI nanoparticles. It is possible to facilitate the design of CS–PPI nanoparticles for stabilizing emulsions and promoting their application in foods, cosmetics, pharmaceuticals and other fields.

## 2. Materials and Methods

### 2.1. Materials

Chitosan (CS, degree of deacetylation 80.0~95.0%) was obtained from Sinopharm chemical reagent Co., Ltd. (Shanghai, China). Pea protein isolate (PPI, protein content ≥ 85%) was obtained from Yosin Biotechnology (Yantai) Co., Ltd. (Shandong, China). Acetic acid and hydrochloric acid were purchased from Yantai Sanhe Chemical Reagent Co., Ltd. (Shandong, China). Dodecyl Sodium Sulfonate (97% purity) was acquired by Shanghai Macklin Biochemical Co., Ltd. (Shandong, China). NaOH was obtained from Sinopharm chemical reagent Co., Ltd. (Shanghai, China). Medium-chain triglyceride (MCT) oil was purchased from Shanghai Yuanye Bio-Technology Co., Ltd. (Shanghai, China). Corn oil was supplied by Longyuan Oil Food Co., Ltd. (Shandong, China).

### 2.2. CS–PPI Nanoparticles Preparation

The method of preparation of CS–PPI nanoparticles was based on the method reported by Ji et al. (2022) [15], with slight modifications. PPI (2.0%, *w/v*) was dissolved in water with magnetic stirring for 30 min. The pH of the solution was adjusted to 9.5 with NaOH aqueous solution (1 M) and stirred for 2 h. Then, the pH was adjusted to 7.0 with HCl aqueous solution (0.1 M) and stirred for 3 h. Finally, the PPI solution was centrifuged at 6000 rpm for 15 min to remove any insoluble substance. CS (0.5%, *w/v*) was added to acetic acid (1.0%, *w/v*) with magnetic stirring until there are no large particles and then solubilized with 20 min sonication at 25°C. Next, the solution was filtered with a 0.45 μm pore-size filtering membrane. The nanoparticles were prepared by CS solution and PPI solution with volume ratios (1:1, 1:2, 1:3, 1:4, 4:1, 3:1 and 2:1) with 500 rpm stirring for 1 h until forming a white suspension (CS–PPI nanoparticles). The suspension was freeze-dried for 2 days by an Alpha 1-2 LD plus vacuum freeze dryer (Marin Christ Corporation, Osterode, Germany) and stored at −20 °C.

### 2.3. Particle Size, Polydispersity Index and Zeta Potential Determination

The particle size, polydispersity index (PDI) and zeta potential of CS–PPI nanoparticles of all ratios, CS and PPI were measured using a NanoBrook 90Plus nanoparticle size analyzer (Brookhaven Instruments Corporation, Beijing, China). Three different ratios of CS–PPI nanoparticles were diluted 20 times with deionized water, mixing the dispersion well at every dilution. Samples were placed in the measurement cell and analyzed at 25 °C. Each sample was tested in triplicate.

### 2.4. Zeta Potential of CS–PPI Nanoparticles at Different pH

CS–PPI nanoparticles with ratios of 1:1, 1:2, 1:3 and PPI solution were diluted with deionized water to a PPI content of 0.5% (*w/v*). Then CS–PPI nanoparticles with three volume ratios, PPI solution, and CS solution were adjusted from 4.0 to 7.5 pH using NaOH (1 mol/L) and HCl (0.1 mol/L), with each increase of 0.5 pH as a unit. The potential of each sample was measured with the NanoBrook 90Plus nanoparticle size analyzer (Brookhaven Instruments Corporation, Beijing, China), the measurement temperature was 25 °C, and measurements were in triplicate.

### 2.5. Turbidity

CS–PPI nanoparticles solutions, PPI solution and CS solution were prepared, as described in 2.4. The absorbance of each sample was measured at 600 nm with a UV-6100 spectrophotometer (Shanghai Metash Instruments Co. Ltd., Shanghai, China). Deionized water was used as a blank. Taking pH value as the independent variable and absorbance value (OD_600_) as the dependent variable, the turbidity change curve was drawn.

### 2.6. Endogenous Fluorescence Spectroscopy

The fluorescence of three different ratios of CS–PPI nanoparticles, CS and PPI was determined by a RF-6000 fluorescence spectrophotometer (Shimadzu (China) Co. Ltd., Shanghai, China). Before the experiment, three different ratios of CS–PPI nanoparticles and PPI were diluted with deionized water to a final protein concentration of 0.05%. The excitation wavelength was set at 285 nm, and the emission spectra were collected between 300 and 450 nm with a scanning speed of 200 nm/min. The data interval was 0.5nm. Each sample was tested in triplicate.

### 2.7. Fourier Transform Infrared Spectroscopy (FTIR)

CS–PPI nanoparticles of three different ratios, CS and PPI were analyzed using a Frontier infrared spectrometer (PerkinElmer Management (Shanghai) Co. Ltd., Shanghai, China). The FTIR spectra were recorded in absorbance mode from 4000 to 400 cm^−1^ after background subtraction.

### 2.8. Interface Properties

#### 2.8.1. Wettability

The three-phase contact angle (*θ_O/W_*) was measured using a CA1000C Contact Angle Goniometer (Shanghai Innuo Precision Instruments Co. Ltd., Shanghai, China). Briefly, freeze-dried samples of PPI, CS and CS–PPI nanoparticles in three ratios were compressed onto a thin film. This protein film was then immersed in the MCT, which was in a transparent container. Deionized water (50 μL) was gently placed on the surface of the tablets using a high-precision injector. After equilibrium was reached, the droplets were photographed and the contact angles were determined.

#### 2.8.2. Emulsifying Properties

The emulsifying activity and emulsifying stability of CS–PPI nanoparticles and PPI solution were determined according to a turbidimetric method [16], with some modifications. All samples were diluted with deionized water to a final protein concentration of 0.05% (*w/v*). The emulsion, from 2.0 mL of corn oil and 8.0 mL of sample diluent, was homogenized with a T18 high-speed dispersing machine (IKA Works Guangzhou, Guangzhou, China) at 15,000 rpm for 2 min. One hundred microliters of emulsion were taken immediately (0 min) from the homogenized emulsion and added into 14.90 mL of 0.1% (*w/v*) SDS solution. Then, the absorbance of diluted emulsions was measured at 500nm on a TU-1900 visible spectrophotometer (Beijing Purkinje GENERAL Instrument Co., Ltd., Beijing, China). This test was repeated after 10min with the same volume of sample. Each sample was tested in triplicate. Emulsifying activity index (EAI) and emulsifying stability (ESI) were calculated using Equations (1) and (2), respectively:(1)$$\mathrm{EAI}\left(m^{2}/g\right)=\frac{2\times 2.303\times A_{0}\times D}{c\times \varphi \times 10^{4}}$$
(2)$$\mathrm{ESI}\left(\%\right)=\frac{A_{10}}{A_{0}}\times 100$$
here, *D* is the dilution factor, *c* is the initial concentration of protein (g/mL), *φ* is the volume fraction of oil in the emulsion (*v*/*v*) (*φ* = 0.2), *A_0_* is the initial absorbance (0 min), and *A_10_* is the absorbance at 10 min after homogenization.

### 2.9. Thermal Stability

The thermal stability of three different ratios of CS–PPI nanoparticles was investigated using a NanoBrook 90Plus nanoparticle size analyzer (Brookhaven Instruments Corporation, Beijing, China), Observing the change of particle size before and after heating. CS–PPI nanoparticle solutions of three different ratios were heated in a water bath at 70 °C for 0, 5, 10, 20, 30 and 60 min, and then immediately put into water for rapid cooling. Before measurement, the solutions were diluted 20 times with deionized water and the dispersion was mixed with each dilution. Each sample was tested in triplicate.

### 2.10. Transmission Electron Microscopy (TEM)

The morphology of the freshly prepared CS–PPI nanoparticles of three ratios was determined with a JEM-1400plus transmission electron microscope (JEOL Ltd., Tokyo, Japan). A drop of CS–PPI nanoparticle solution was placed on a copper mesh and allowed to air dry.

### 2.11. Statistical Analysis

All experiments were conducted in triplicate, and data were presented as average values. The statistical significance was determined by one-way analysis of variance and Duncan’s multiple range test using SPSS 22.0 and *p* < 0.05 was considered statistically significant.

## 3. Results and Discussion

### 3.1. Particle Size, PDI and Zeta Potential

The particle size, PDI and zeta potential of CS, PPI, and CS–PPI nanoparticles were shown in Table 1. The particle size of CS–PPI nanoparticles with volume ratios of 1:1, 1:2, and 1:3 was nanoscale, with the smallest particle size among seven ratios with no significant differences. The particle size of CS–PPI nanoparticles with a volume ratio of 1:4 increased significantly with the increase in protein concentration, which may be due to the aggregation of the protein itself. In addition, the PDI of CS–PPI nanoparticle solutions, PPI solution and CS solution were all about 0.3. There was no significant difference among CS–PPI complexes (*p* < 0.05). This indicated that the diluted particles had a better dispersion [17]. As for zeta potential, the potential of the complexes was around or over 30 mV, suggesting that the complexes are relatively stable and dispersed in the solution [18]. It can be concluded that the concentration of chitosan plays an important role in the potential of the CS–PPI nanoparticle solution. The potential of the nanoparticle dispersion became larger when CS accounted for the larger proportion. The possible reason for this is that the potential of the system gradually increases because of the increase in chitosan concentration. Part of the CS was attached or bound to the PPI. The excess chitosan was free in the mixing due to electrostatic repulsion resulting in an increase in potential in the system.

### 3.2. Zeta Potential of CS–PPI Nanoparticles at Different pH

It is important to know the potentials of PPI, CS and CS–PPI nanoparticles with three different ratios at different values of pH, which helps analyze the binding mode between pea protein isolate and chitosan. As shown in Figure 1A, the CS solution is positively charged under acidic conditions, and the isoelectric point of PPI is around 3. The potential of CS–PPI nanoparticle solutions decreased with the increase in pH. The potential decreased rapidly when the pH was greater than 5.5, indicating that the pH value can greatly affect the surface charge of CS–PPI nanoparticle solutions. Among them, the solution of CS and PPI at a volume ratio of 1:1 showed the slowest downward trend, showing that the mixing ratio of protein and polysaccharide also affects the balance of the charge of the complexes. PPI had a negative charge when the pH increased from 4.0 to 8.0, while CS had a positive charge. Thus, CS and PPI were mainly combined by electrostatic interaction [19]. The pH of the original CS–PPI nanoparticle solution was about 4.3. CS was positively charged; PPI was negatively charged. It showed that CS and PPI can interact electrostatically to form CS–PPI nanoparticles.

### 3.3. Turbidity

The strength of electrostatic interaction can be controlled by pH since it can influence the charge of protein [20]. Therefore, proteins and polysaccharides can form soluble and insoluble complexes through electrostatic interactions in a specific pH range, with a critical pH transition point (pH_S_) as a demarcation point [9]. In this experiment, CS and PPI will form a soluble complex when the pH of the solution is less than pH_S_. On the contrary, an insoluble complex is formed when the pH is greater than pH_S_.

As shown in Figure 1B, changes in the optical density (OD_600_) of CS, PPI and three ratios of CS–PPI nanoparticles at pH 3–8 were measured. The optical density of the PPI solution remained relatively high at pH 4~6. When the pH is greater than 6.2, the optical density decreased and tended to be stable. As for CS, the optical density of CS was zero when the pH was less than 6.0. Then it increased sharply and reached the final maximum value at a pH value of 7.0. The reason for the increase in optical density of the chitosan solution is the deprotonation of the amino groups of chitosan as the pH becomes greater than its pKa (pH 6.0), which reduced the electrostatic charge on the molecular surface [21]. The intermolecular repulsion was weakened, causing their aggregation. The solubility of chitosan decreased and its turbidity increased. This conclusion corresponds to the results in Figure 1A, the potential of chitosan gradually decreased when the pH was greater than 6.0. The critical pH value (pH_S_) for the range of nanoparticles (1:1, 1:2, 1:3) was 6.5, 5.6, 6.0, respectively. CS and PPI soluble complexes formed under the critical pH value (pH_S_). The optical densities of the three complexes were quite different from those of CS and PPI alone. It indicated the existence of intermolecular forces between CS and PPI. In addition, the mixing ratio of protein and polysaccharide also affected the charge balance in the complex, resulting in the difference in its critical pH value (pH_S_). The optical density of the range of nanoparticles reached a maximum at pH 7, 6.5, and 7, respectively. At this time, the surface charge of the nanoparticles was close to zero, as in Figure 1A. The intermolecular electrostatic repulsion is the smallest and aggregation is likely to occur between the molecules. It is consistent with the conclusion of Yang et al. (2020) [22] and Elmer et al. (2011) [23]. Lan et al. (2018) stated that the formation of insoluble complexes results from the greater attraction between PPI and HMP [9]. The potential difference between PPI and HMP almost reached a maximum of around pH 7. The point of maximum optical density almost always appeared around pH 7. The different positions are due to the different proportions of CS and PPI.

### 3.4. Endogenous Fluorescence Spectroscopy

The above research results have shown that CS and PPI can form nanoparticles according to a certain proportion and within a certain range of conditions. Endogenous fluorescence spectroscopy is particularly sensitive to the microenvironment of protein tryptophan residues. It is often used as a means to detect protein spatial conformational changes. Therefore, endogenous fluorescence spectroscopic analysis was performed in order to further study the complex behavior between the two.

The fluorescence spectra of CS–PPI nanoparticles with different mixing ratios, and CS and PPI solutions at the excitation wavelength of 285 nm were shown in Figure 2A. Chitosan had almost no absorption peak in the range of 300–450 nm. The maximum fluorescence emission wavelength of all CS–PPI nanoparticles is redshifted from 322.0 nm to 325.5 nm for PPI. A slight redshift is sufficient to indicate that the hydrophobic amino acids within the pea protein isolate are exposed to more polar solvents, which in turn indicate the unfolding of the PPI tertiary structure [24,25]. It may be because the combination of positively charged chitosan and negatively charged pea protein isolate through electrostatic attraction induces the unfolding of PPI, exposing the hydrophobic amino acids to a more polar aqueous environment [26]. Besides, it can be seen that the fluorescence intensities of PPI, nanoparticles with the ratio of 1:2, 1:3, and 1:1 decreased from 35,139 to 33,985, 33,891 and 33,379 a.u., respectively. It can be concluded that the fluorescence intensity of nanoparticles decreased when the proportion of chitosan increased, which was consistent with the result of Yi et al. (2020) [27]. The decrease in fluorescence intensity indicated that the fluorescent group of the hydrophobic amino acid of PPI was quenched. The redshift of the absorption wavelength of the highest fluorescence intensity and the decrease in the fluorescence intensity indicated the complexation of CS and PPI [27,28]. Combined with Figure 1A, it can be concluded that the complexation of CS and PPI may be caused by electrostatic interaction.

### 3.5. Fourier Transform Infrared Spectroscopy (FTIR) Spectra

Fourier transform infrared spectroscopy can be used to analyze the interaction between functional groups in complexes [29]. Figure 2B displays the FTIR spectra of CS, PPI and CS–PPI nanoparticles. Overall, the four characteristic peaks of PPI were 3287, 2961, 1655 and 1546 cm^−1^, which are respectively assigned to the stretching vibration of OH groups, C-H groups, the amide I band (C=O stretching) and amide II bands (C-N stretching and N-H stretching) [30]. For CS, the absorption peaks at 3412 cm^−1^ represented a strong amino characteristic peak. The peaks at 1632 and 1560 cm^−1^ were attributed to the amide I band (C=O vibration) and the amide III band (-NH^3+^), respectively. Moreover, the peak at 1409 cm^−1^ referred to the OH and C-H vibrations, the peak at 1153 cm^−1^ was related to the symmetrical stretching of C-O-C, and the peak at 1087 cm^−1^ corresponded to the C-O stretching vibration [31].

Three different ratios of nanoparticles were compared with CS and PPI, respectively. The two peaks at 1655 cm^−1^ of PPI and 1560 cm^−1^ of CS did not appear in the nanoparticles. They represented the carboxyl group and amino group, respectively. Instead, they were replaced by the two peaks of 1645 and 1554 cm^−1^ in the nanoparticle (1:1), two peaks of 1646 and 1548 cm^−1^ in the nanoparticle (1:2), two peaks of 1649 and 1548 cm^−1^ in the nanoparticle (1:3), which indicated that the appearance of the new peak was generated by the combination of carboxyl and amino groups through electrostatic attraction [31]. The peak representing the OH group in PPI (3287 cm^−1^) experienced a redshift in all three nanoparticles. Further, there were more redshifted wave numbers with a larger proportion of CS. The redshifted wavelengths of nanoparticles (1:1, 1:2, 1:3) were 136, 128 and 104 cm^−1^, respectively, which suggests that hydrogen bonds are involved in the formation of nanoparticles [32], and the redshift is related to the concentration of chitosan.

### 3.6. Interface Properties

#### 3.6.1. Wettability

Solid particles with better interfacial wettability can promote the stability of Pickering emulsions. The wettability of particles is usually determined by the contact angle [33]. When the contact angle of solid particles is close to 90°, the adsorption of effective particles on the oil/water interface can be promoted, and the aggregation of oil droplets can also be sterically hindered [34]. The wetting properties of the particles were measured by investigating the three-phase contact angles (*θ_O/W_*) of CS–PPI nanoparticles, CS and PPI, which were immersed in MCT. Images of water drops deposited onto sample tablets immersed in MCT are shown in Figure 3. It could be found that the contact angle (*θ_O/W_*) of pure CS was 121°, indicating that it was quite hydrophobic and tended to be more lipophilic in emulsions. On the contrary, pure PPI was found to have a contact angle of 64.2°. It indicated that it was hydrophilic. As shown in Figure 3, the contact angle of the nanoparticles was between CS and PPI. The decrease in the *θ_O/W_* of CS–PPI nanoparticles proved the successful surface modification of overly hydrophobic CS with hydrophilic PPI [35]. It was suggested that the combination of the two could influence the contact angle by changing the group’s composition on the surface of the complexes. It was worth noting that the hydrophilicity of the nanoparticles increased with the increase in the proportion of PPI. The contact angle of the nanoparticle (1:3) was closest to 90°.

#### 3.6.2. Emulsifying Properties

The emulsifying activity index (EAI) represents the ability of proteins to be adsorbed at the interface. The emulsifying stability index (ESI) represents the protein capacity for staying at the water-oil interface after emulsion storage or heating [36]. It can be seen from Figure 4A that the emulsifying activity (EAI) of CS–PPI nanoparticles improved with the increase in the proportion of PPI. PPI plays a decisive role in the emulsifying activity. This result validates the results for the contact angle. The results of emulsion stability (ESI) are shown in Figure 4B, nanoparticles with a high proportion of chitosan (1:1) had better emulsion stability than other proportions of nanoparticles and pure PPI. This may be related to the potential of CS–PPI nanoparticles [37]. The overall potential of the nanoparticles was increased due to the chitosan. Nanoparticles with higher potentials are more stable in emulsions.

### 3.7. Thermal Stability

The denaturation temperature of pea protein isolate is 75 °C and above [38]. For this reason, 70 °C was used as the heating stability. The structure of the protein was not destroyed under this temperature. The thermal stability of CS–PPI nanoparticles is judged by measuring the particle size change after heating. As shown in Figure 5, the particle size of nanoparticles decreased with the increase in heating time and remained basically unchanged after 20 min. The particle size reduction of 1:1, 1:2, and 1:3 nanoparticles is approximately 106, 137, and 127 nm, respectively. It can be seen that the thermal stability of nanoparticles from high to low is 1:1, 1:3, 1:2. The particle size decreased with the increase in heating time, which may be due to the depolymerization of aggregated chitosan molecules and the degradation of chitosan molecular chains.

### 3.8. Transmission Electron Microscopy (TEM)

Three different ratios of CS–PPI nanoparticles were displayed by transmission electron microscopy, and the observation results were shown in Figure 6. When the ratio of CS to PPI is 1:1, a lump similar to a woolen ball was formed, and the edge was blurred. Another possibility for this morphogenesis is that the particles were influenced by the e-beam [39]. As for when the ratio of CS to PPI was 1:2, there was a branched chain structure. When the ratio of CS to PPI was 1:3, spherical nanoparticles were formed with clear edges. It could be seen that the ratio of pea protein isolate and chitosan played an important role in the formation of spherical nanoparticles.

## 4. Conclusions

In this study, the structural and functional properties of CS–PPI nanoparticles were evaluated. It can be concluded that CS and PPI are mainly bound by electrostatic interaction and hydrogen bonding, as seen from the results of the potential change, turbidity, endogenous fluorescence spectroscopy and FTIR spectra. Nanoparticle (1:3) had the best interface property according to the results of the contact angles and the EAI. Furthermore, it can be seen from the thermal stability that the addition of CS improved the thermal stability of CS–PPI nanoparticles. The morphology of the CS–PPI nanoparticles was close to spherical according to TEM images. This study provides a theoretical basis for the targeted design of CS–PPI nanoparticles, which will help to explore the mechanism of CS–PPI nanoparticles stabilizing Pickering emulsions.

## Figures and Tables

**Figure 1 molecules-27-06913-f001:**
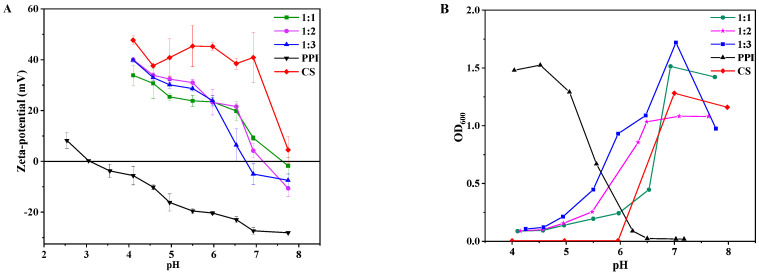
Zeta potential (**A**) and turbidity curves (**B**) of the CS–PPI nanoparticles solutions, PPI solution and CS solution at different pH values.

**Figure 2 molecules-27-06913-f002:**
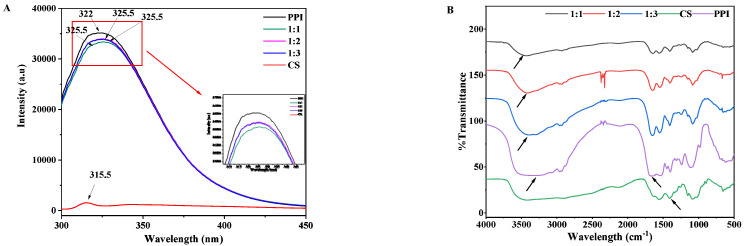
Fluorescence emission spectra (**A**) and FTIR spectra (**B**) of the CS–PPI nanoparticles, CS and PPI.

**Figure 3 molecules-27-06913-f003:**
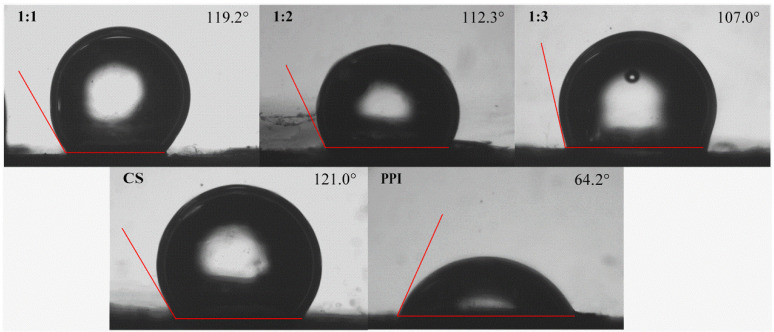
Oil-in-water three-phase contact angles (*θ_O/W_*) of CS–PPI nanoparticles.

**Figure 4 molecules-27-06913-f004:**
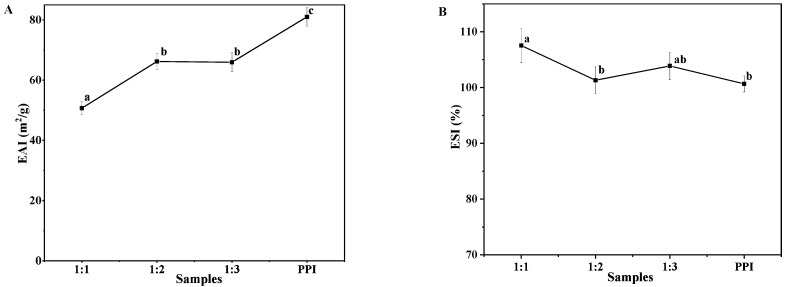
Emulsifying activity index (EAI) (**A**) and emulsifying stability index (ESI) (**B**) of PPI and CS–PPI nanoparticles with different ratios. Different superscript letters in the figure mean significant differences (*p* < 0.05).

**Figure 5 molecules-27-06913-f005:**
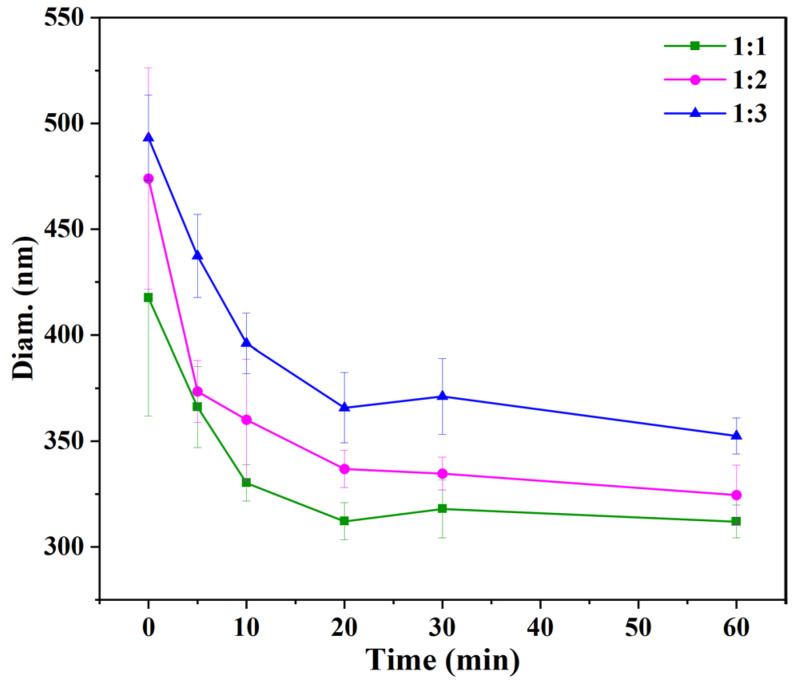
Effect of heat-treatment time on the particle size of CS–PPI nanoparticles with different ratios.

**Figure 6 molecules-27-06913-f006:**
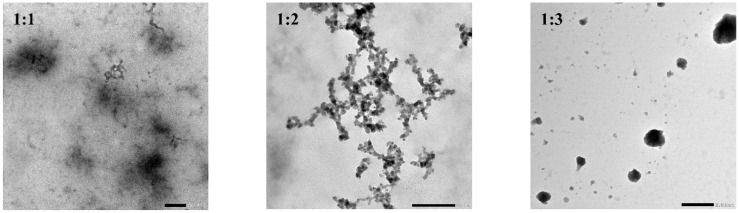
Transmission electron microscopy images of CS–PPI nanoparticles.

**Table 1 molecules-27-06913-t001:** Particle size, PDI and zeta potential of samples.

Samples	Effective Diameter (nm)	PDI	Zeta Potential (mV)
PPI	229.32 ± 4.59 ^d^	0.340 ± 0.013 ^ab^	−15.86 ± 0.64 ^f^
CS	786.06 ± 10.80 ^c^	0.376 ± 0.081 ^a^	38.77 ± 0.53 ^b^
1:1	802.95 ± 71.94 ^bc^	0.337 ± 0.061 ^ab^	45.74 ± 0.56 ^a^
1:2	807.10 ± 86.22 ^c^	0.308 ± 0.040 ^ab^	36.82 ± 0.59 ^c^
1:3	767.75 ± 110.10 ^c^	0.311 ± 0.045 ^ab^	27.58 ± 1.04 ^d^
1:4	1108.11 ± 280.50 ^a^	0.299 ± 0.021 ^b^	20.58 ± 1.31 ^e^
4:1	1042.15 ± 118.00 ^ab^	0.330 ± 0.036 ^ab^	46.84 ± 1.11 ^a^
3:1	962.76 ± 68.05 ^ac^	0.348 ± 0.012 ^ab^	45.75 ± 0.89 ^a^
2:1	1119.72 ± 95.84 ^a^	0.311 ± 0.039 ^ab^	46.26 ± 0.95 ^a^

Different lowercase letters (a–f) in the same column indicated significant differences (*p* < 0.05).

## Data Availability

Not applicable.

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
