# Peer review of "Fabrication and Characterization of Chitosan–Pea Protein Isolate Nanoparticles"

_molecules, 2022, doi:10.3390/molecules27206913_

Round 1
Reviewer 1 Report
In general the paper is interesting and does provide some new data on the properties of pea protein/chitosan mixtures. However as it stands the paper does not really reach the standard required for publication. The abstract should clearly state the contact angle for all 3 systems studied- not just the 1:3 ratio sample. In the first line of the Introduction the phrase 'kind of' should be deleted. This is un-scientific language. In the Results section mv should be replaced with mV. In section 3.3 the authors should note that pH can also greatly influence chitosan via potential protonation of the amine groups and not just the charge on the pea protein. To ignore this is to be too simplistic. Also in this section the authors must explain what they mean by the phrase 'A large number of PPI and CS aggregated.' In section 3.4 do the authors really think a red shift of 3.5 nm is really due to conformational change in the protein? In my opinion this is a very unlikely scenario. The authors need to greatly strengthen their argument here with more references in order to make this conclusion sound plausible. The spectra presented look identical to me. What specific transitions do the spectra represent? This is not described. In section 3.5 it seems very dubious to me that the authors can quote band positions to 0.01 cm-1. This must be specifically justified as it again seems very unlikely. In section 3.8 I find the electron microscopy results to be indicative of e-beam damage. The image for the 1:1 sample shows no evidence of nanoparticle formation. The charging effect of the e-beam will cause damage and agglomeration. The authors must include remarks on these possibilities. The overall conclusion that the interaction between the protein and the chitosan is most likely via just H-bonding is probably correct, but the accompanying conclusions are not really justified.
Author Response
Point 1: In general, the paper is interesting and does provide some new data on the properties of pea protein/chitosan mixtures. However, as it stands the paper does not really reach the standard required for publication.
Response 1: We appreciate the selfless contributions to this manuscript from you. It is because of your insightful and helpful comments and suggestions, we gained the confidence to improve our work better. The manuscript molecules-1924766 entitled “Fabrication and Characterization of Chitosan-Pea Protein Isolate Nanoparticles” has been carefully revised with the revisions marked up using the “Track Changes” function. The point-by-point answers to these comments and suggestions were listed as below.
Point 2: The abstract should clearly state the contact angle for all 3 systems studied-not just the 1:3 ratio sample.
Response 2: Many thanks for your comment. We have revised the abstract as follows: “In terms of interface properties, the contact angles of nanoparticles with the ratio of 1:1, 1:2, 1:3 were 119.2°, 112.3°, 107.0°, respectively.” in lines 22-24 in the revised manuscript.
Point 3: In the first line of the Introduction the phrase 'kind of' should be deleted. This is un-scientific language. In the Results section mv should be replaced with mV.
Response 3: Thank you for your comments. We have deleted the phrase 'kind of' in line 40 and revised “mv” into “mV” in line 250 in the revised manuscript.
Point 4: In section 3.3 the authors should note that pH can also greatly influence chitosan via potential protonation of the amine groups and not just the charge on the pea protein. To ignore this is to be too simplistic.
Response 4: Many thanks for your comment. The charge on the surface of chitosan and proteins both can be influenced by pH of the solution. When the pH in the solution is higher than its pKa (pH 6.0), chitosan will deprotonate and their solubility can be reduced. In order to express the above views more clearly, we supplemented explanation of the deprotonation of chitosan in lines 291-296 in the revised manuscript which were shown as follows: “The reason for the increase in optical density of chitosan solution is the deprotonation of the amino groups of chitosan as the pH is greater than its pKa (pH 6.0), which reduced the electrostatic charge on the molecular surface [21]. The intermolecular repulsion was weakened, causing their aggregation. The solubility of chitosan decreased and its turbidity increased. This conclusion corresponds to the results in Fig. 1(A), the potential of chitosan gradually decreased when the pH was greater than 6.0.”
- Zhang, Q.; Dong, H.M.; Gao, J.; Chen, L.Y.; Vasanthan, T. Field pea protein isolate/chitosan complex coacervates: Formation and characterization. Carbohydrate Polymers 2020, 250, doi:10.1016/j.carbpol.2020.116925.
Point 5: Also in this section the authors must explain what they mean by the phrase 'A large number of PPI and CS aggregated.'
Response 5: Thank you for your comments. When the turbidity of the solution reaches a maximum, it corresponds to the point where the potential of the nanoparticle is zero. At this time, the surface charge of the molecules is zero. The repulsion between the molecules is reduced, and aggregation is likely to occur.
We have supplemented explanation of 'A large number of PPI and CS aggregated' in lines 303-311 in the revised manuscript which was shown as follows “At this time, the surface charge of the nanoparticles was close to zero in Fig. 1(A). The intermolecular electrostatic repulsion is the smallest and aggregation is likely to occur between the molecules. It is consistent with the conclusion of Yang et al. (2020) [22] and Elmer et al. (2011) [23]. Lan et al. (2018) stated that the formation of insoluble complexes results from the greater attraction between PPI and HMP [24]. The potential difference between PPI and HMP almost reached a maximum around pH 7. The point of maximum optical density almost all appeared around pH 7. The different positions are due to the different proportions of CS and PPI.”
- Yang, S.N.; Li, X.F.; Hua, Y.F.; Chen, Y.M.; Kong, X.Z.; Zhang, C.M. Selective Complex Coacervation of Pea Whey Proteins with Chitosan To Purify Main 2S Albumins. Journal of Agricultural and Food Chemistry 2020, 68, 1698-1706, doi:10.1021/acs.jafc.9b06311.
- Elmer, C.; Karaca, A.C.; Low, N.H.; Nickerson, M.T. Complex coacervation in pea protein isolate-chitosan mixtures. Food Research International 2011, 44, 1441-1446, doi:10.1016/j.foodres.2011.03.011.
- Lan, Y.; Chen, B.C.; Rao, J.J. Pea protein isolate-high methoxyl pectin soluble complexes for improving pea protein functionality: Effect of pH, biopolymer ratio and concentrations. Food Hydrocolloids 2018, 80, 245-253, doi:10.1016/j.foodhyd.2018.02.021.
Point 6: In section 3.4 do the authors really think a red shift of 3.5 nm is really due to conformational change in the protein? In my opinion this is a very unlikely scenario. The authors need to greatly strengthen their argument here with more references in order to make this conclusion sound plausible.
Response 6: Many thanks for your comment. We have checked out a certain amount of related papers and found some ideas to prove that the 3.5 nm red shift leads to the change of protein conformation. The further explanations are shown in lines 326-333 in the revised manuscript as follows: “The maximum fluorescence emission wavelength of all CS-PPI nanoparticles is red-shifted from 322.0 nm to 325.5 nm for PPI. A slight red shift is sufficient to indicate that the hydrophobic amino acids within the pea protein isolate are exposed to more polar solvents, which in turn indicate the unfolding of the PPI tertiary structure [25,26]. It may be because the combination of positively charged chitosan and negatively charged pea protein isolate through electrostatic attraction induces the unfolding of PPI, exposing the hydrophobic amino acids to a more polar aqueous environment [27].”
- Chao, D.; Jung, S.; Aluko, R.E. Physicochemical and functional properties of high pressure-treated isolated pea protein. Innovative Food Science & Emerging Technologies 2018, 45, 179-185, doi:10.1016/j.ifset.2017.10.014.
- Zhang, M.; Feng, X.M.; Liang, Y.R.; He, M.Y.; Geng, M.J.; Huang, Y.Y.; Teng, F.; Li, Y. Effects of electron beam irradiation pretreatment on the structural and functional properties of okara protein. Innovative Food Science & Emerging Technologies 2022, 79, doi:10.1016/j.ifset.2022.103049.
- Guo, Q.; Su, J.Q.; Shu, X.; Yuan, F.; Mao, L.K.; Gao, Y.X. Development of high methoxyl pectin-surfactant-pea protein isolate ternary complexes: Fabrication, characterization and delivery of resveratrol. Food chemistry 2020, 321, doi:10.1016/j.foodchem.2020.126706.
Point 7: The spectra presented look identical to me. What specific transitions do the spectra represent? This is not described.
Response 7: Thank you for your suggestion. It can be seen that the fluorescence intensity of the three nanoparticles is smaller than that of PPI. Among the fluorescence intensities of three nanoparticles, the fluorescence intensity of nanoparticle with the ratio of 1:1 was the smallest. These changes reflected the binding of CS to PPI.
In order to express the above views more clearly, we have revised the sentences in lines 333-342 in the revised manuscript which were shown as follows: “Besides, it can be seen that the fluorescence intensities of PPI, nanoparticles with the ratio of 1:2, 1:3, and 1:1 decreased from 35139 to 33985, 33891 and 33379 a.u, respectively. It can be concluded that the fluorescence intensity of nanoparticles decreased when the proportion of chitosan increased, which was consistent with the result of Yi et al. (2020) [28]. The decrease of fluorescence intensity indicated that the fluorescent group of the hydrophobic amino acid of PPI was quenched. The red-shift of the absorption wavelength of the highest fluorescence intensity and the decrease of the fluorescence intensity indicated the complexation of CS and PPI [28,29]. Combined with Fig. 1(A), it can be concluded that the complexation of CS and PPI may be caused by electrostatic interaction.”
- Yi, J.; Gan, C.; Wen, Z.; Fan, Y.T.; Wu, X.L. Development of pea protein and high methoxyl pectin colloidal particles stabilized high internal phase pickering emulsions for beta-carotene protection and delivery. Food Hydrocolloids 2021, 113, doi:10.1016/j.foodhyd.2020.106497.
- Wang, Q.; Pan, M.H.; Chiou, Y.S.; Li, Z.S.; Wei, S.D.; Yin, X.L.; Ding, B.M. Insights from alpha-Lactoalbumin and beta-Lactoglobulin into mechanisms of nanoliposome-whey protein interactions. Food Hydrocolloids 2022, 125, doi:10.1016/j.foodhyd.2021.107436.
Point 8: In section 3.5 it seems very dubious to me that the authors can quote band positions to 0.01 cm-1. This must be specifically justified as it again seems very unlikely.
Response 8: Thanks for your critical suggestion. The result of 0.01 cm-1 was found using OMNIC. Now I know this expression is unreasonable. In fact, the accuracy of the FTIR spectrometer we used was 1 cm-1. We have checked the whole manuscript carefully and revised the sentences in lines 347-418 in the revised manuscript.
Please see: “Fourier transform infrared spectroscopy can be used to analyze the interaction between functional groups in complexes [30]. Fig. 2(B) displays the FTIR spectra of CS, PPI and CS-PPI nanoparticles. Overall, the four characteristic peaks of PPI were 3287, 2961, 1655 and 1546 cm-1, which are respectively assigned to the stretching vibration of OH groups, C-H groups, the amide I band (C=O stretching) and amide II bands (C-N stretching and N-H stretching) [31]. For CS, the absorption peaks at 3412 cm−1 represented a strong amino characteristic peak. The peaks at 1632 and 1560 cm−1 were attributed to the amide I band (C=O vibration) and the amide III band (-NH3+), respectively. Moreover, the peak at 1409 cm−1 referred to the OH and C-H vibrations, the peak at 1153 cm−1 was related to the symmetrical stretching of C-O-C, and the peak at 1087 cm−1 corresponded to the C-O stretching vibration [32].
Three different ratios of nanoparticles were compared with CS and PPI, respectively. The two peaks at 1655 cm-1 of PPI and 1560 cm-1 of CS did not appear in the nanoparticles. They represented the carboxyl group and amino group, respectively. Instead, they were replaced by the two peaks of 1645 and 1554 cm-1 in the nanoparticle (1:1), two peaks of 1646 and 1548 cm-1 in the nanoparticle (1:2), two peaks of 1649 and 1548 cm-1 in the nanoparticle (1:3), which indicated that the appearance of the new peak was generated by the combination of carboxyl and amino groups through electrostatic at-traction [32]. The peak representing the OH group in PPI (3287 cm-1) experienced a red shift in all three nanoparticles. Besides, there were more red-shifted wave numbers with the larger proportion of CS. The red-shifted wavelengths of nanoparticles (1:1, 1:2, 1:3) were 136, 128 and 104 cm-1, respectively, which suggests that hydrogen bonds are involved in the formation of nanoparticles [33], and the red shift is related to the concentration of chitosan.”
Point 9: In section 3.8 I find the electron microscopy results to be indicative of e-beam damage. The image for the 1:1 sample shows no evidence of nanoparticle formation. The charging effect of the e-beam will cause damage and agglomeration. The authors must include remarks on these possibilities.
Response 9: Thank you for your comments. The point you made helps us understand TEM imaging. We have added the influence of e-beam to the revised manuscript. “Another possibility for this morphogenesis is that the particles were influenced by the e-beam [40].” was added in lines 531-533 in the revised manuscript.
- Turner, E.M.; Sapkota, K.R.; Hatem, C.; Lu, P.; Wang, G.T.; Jones, K.S. Wet-chemical etching of FIB lift-out TEM lamellae for damage-free analysis of 3-D nanostructures. Ultramicroscopy 2020, 216, 113049, doi:https://doi.org/10.1016/j.ultramic.2020.113049.
Point 10: The overall conclusion that the interaction between the protein and the chitosan is most likely via just H-bonding is probably correct, but the accompanying conclusions are not really justified.
Response 10: Many thanks for your comment. In order to make the conclusion of the article more logical, we have revised the conclusion. The modified conclusion was shown in lines 540-549 in the revised manuscript as follows: “In this study, the structural and functional properties of CS-PPI nanoparticles were evaluated. It can be concluded that CS and PPI are mainly bound by electrostatic interaction and hydrogen bonding from the results of the potential change, turbidity, endogenous fluorescence spectroscopy and FTIR spectra. Nanoparticle (1:3) had the best interface property according to the results of contact angles and EAI. Besides, it can be seen from the thermal stability that the addition of CS improved the thermal stability of CS-PPI nanoparticles. The morphology of the CS-PPI nanoparticles was close to spherical according to TEM images. This study provides a theoretical basis for the targeted design of CS-PPI nanoparticles, which will help to explore the mechanism of CS-PPI nanoparticles stabilizing Pickering emulsion.”

Reviewer 2 Report
This research uses CS and PPI to prepare nanoparticles for stabilizing emulsions and promoting their application in food and other fields. It was noted that the PPI solution was centrifuged at 6000 rpm for 15 min to remove any insoluble substance in this study. To my knowledge, the commercial PPI had low solubility at neutral pH, maybe more than half of the PPI proteins were lost. As shown in Fig.1, after centrifuging, the soluble PPI had higher stability than CS-PPI at natural pH due to the PI of the complex moving to around pH7 -8. In this case, PPI also showed better EAI compared with CS-PPI as shown in Fig.4A. Although the author demonstrated that the ESI of the CS-PPI complex was higher than PPI (FIg.4B), the result was not significantly different in my opinion. Moreover, because the author aimed to use CS-PPI nanoparticles to stabilize the emulsion, the changes in emulsion sizes after emulsion preparation and during storage should be observed. In all, it‘s a pity for this research to give no new information on the potential application of the CS-PPI complex.
Author Response
We appreciate the selfless contributions to this manuscript from you. It is because of your insightful and helpful comments and suggestions, we gained the confidence to improve our work better. The manuscript molecules-1924766 entitled “Fabrication and Characterization of Chitosan-Pea Protein Isolate Nanoparticles” has been carefully revised with the revisions marked up using the “Track Changes” function. The point-by-point answers to these comments and suggestions were listed as below.
Point 1: Although the author demonstrated that the ESI of the CS-PPI complex was higher than PPI (Fig.4B), the result was not significantly different in my opinion.
Response 1: Thank you very much for your comments. Through SPSS analysis, it can find that the ESI of the nanoparticle with the ratio of 1:1 is significantly different from the nanoparticle with the ratio of 1:2 and PPI. And the significant difference was marked in Fig. 4(B). We have added “Different superscript letters in the figure mean significant differences (p<0.05).” in lines 513-514 in the revised manuscript.
Point 2: Moreover, because the author aimed to use CS-PPI nanoparticles to stabilize the emulsion, the changes in emulsion sizes after emulsion preparation and during storage should be observed. In all, it’s a pity for this research to give no new information on the potential application of the CS-PPI complex.
Response 2: Thanks for your comments. In this study, we explored the effects of different CS:PPI ratios on the structural and functional properties of nanoparticles. Finally, we found that the interface property of nanoparticle (1:3) was the best according to the results of contact angles and EAI. This study will provide a theoretical basis for the targeted design of CS-PPI nanoparticle. In the future, CS-PPI nanoparticles can be potentially used in the stabilization of Pickering emulsion to gain a deeper understanding of the stabilizing mechanism.

Reviewer 3 Report
Ref. No.: molecules-1924766-peer-review-v1
Title: Fabrication and Characterization of Chitosan-Pea Protein Isolate Nanoparticles
Chanchan Sun, Man Zhang, Zikun Li, Mengqi Dai, Hongjun He, Bin Liang, Xiulian Li and Changjian Ji
In this paper, pea protein isolate (PPI) and chitosan (CS) were used as raw materials to prepare nano-particles. The structures and functional properties of the nanoparticles with three ratios (1:1, 1:2 1:3, CS: PPI) were evaluated. Results provides a theoretical basis for the design of CS-PPI nanoparticles and their applications in promoting emulsion stabilization and delivery of active substances using emulsions.
There are several papers published indicated that pea protein isolate (PPI) and chitosan (CS) can be use as raw materials to prepare nano-particles.
I found the paper to be overall well written and much of it was well described.
Title reflects the content of paper.
Authors used adequate, up-to-date literature but some references were not listed adequately.
The experiment is clearly set up, and the analysis methods, their order, as well as description are mainly understandable.
There are a few corrections need to be done:
- In the title of the paper and throughout the paper, nanoparticles are defined as chitosan-pea protein isolate (CS-PPI). However, in the first sentence of the abstract, as well as in the Introduction, pea-protein isolate was mentioned first, then chitosan. Due to unifying the paper and transparency, if it does not disturb the concept of the paper, the proposal is to harmonize the name and abbreviation with the introduction of macromolecules (CS and PPI) into the paper.
- In Material and methods, in pars 2.4 and 2.5 there is no need to duplicate the sentences “SC-PPI nanoparticles with ratio… Then CS-PPI nanoparticles with three…”. It is possible to refer to the statement from part 2.4 without repetition in part 2.5.
- Line 131: is it correct “PPI solution” or it should be “PPI solution”?
- Abbreviation PDI in not introduced in Material and methods.
- In the list of references, several correction is needed – some journals were written in abbreviation; in some cases, journal LWT was written as Lwt and somewhere else Lwt-Food Science and Technology.
Author Response
We appreciate the selfless contributions to this manuscript from you. It is because of your insightful and helpful comments and suggestions, we gained the confidence to improve our work better. The manuscript molecules-1924766 entitled “Fabrication and Characterization of Chitosan-Pea Protein Isolate Nanoparticles” has been carefully revised with the revisions marked up using the “Track Changes” function. The point-by-point answers to these comments and suggestions were listed as below.
Point 1: Authors used adequate, up-to-date literature but some references were not listed adequately.
Response 1: Thanks for your comments. Based on your suggestions, we have made some changes of result analysis and supplemented some appropriate literatures in the revised manuscript to make the conclusion more reasonable. Please see lines 291-293, 305-308, 328-340, 531-533 in the revised manuscript as follows:
“The reason for the increase in optical density of chitosan solution is the deprotonation of the amino groups of chitosan as the pH is greater than its pKa (pH 6.0), which reduced the electrostatic charge on the molecular surface [21]”
“It is consistent with the conclusion drawn by Yang et al. (2020) [22] and Elmer et al. (2011) [23]. Lan et al. (2018) stated that the formation of insoluble complexes results from the greater attraction between PPI and HMP [24].”
“A slight red shift is sufficient to indicate that the hydrophobic amino acids within the pea protein isolate are exposed to more polar solvents, which in turn indicate the unfolding of the PPI tertiary structure [25,26]. …The decrease of fluorescence intensity indicated that the fluorescent group of the hydrophobic amino acid of PPI was quenched. The red-shift of the absorption wavelength of the highest fluorescence intensity and the decrease of the fluorescence intensity indicate the complexation of CS and PPI [28,29].”
“Another possibility for this morphogenesis is that the particles were influenced by the e-beam [40].”
Point 2: In the title of the paper and throughout the paper, nanoparticles are defined as chitosan-pea protein isolate (CS-PPI). However, in the first sentence of the abstract, as well as in the Introduction, pea-protein isolate was mentioned first, then chitosan. Due to unifying the paper and transparency, if it does not disturb the concept of the paper, the proposal is to harmonize the name and abbreviation with the introduction of macromolecules (CS and PPI) into the paper.
Response 2: Many thanks for your comment. In order to unify the paper, we have changed the order of pea protein isolate and chitosan in the abstract and introduction. We revised “Pea protein isolate (PPI) and chitosan (CS)” into “Chitosan (CS) and pea protein isolate (PPI)” in line 16 in the revised manuscript. In addition, we have revised the introduction order of PPI and CS in lines 34-85 in the revised manuscript, which was shown as: “Chitosan (CS) is the only cationic polysaccharide in nature and exists in the deacetylated form of chitin. It has great biodegradability, biocompatibility, and is environmentally friendly [1]. Due to its unique charge, CS can bind to negatively charged proteins through electrostatic interactions in a specific pH range [2]. In addition, CS participated in the encapsulation of active substances can resist the destruction of the strong acidity of gastric juice, which protect the active substances and improve their bioavailability [3].
Pea protein isolate (PPI) is a natural protein, which has certain emulsifying properties, foaming properties and water-holding properties. Besides, compared with soy protein, it has low allergenicity and is safer [4]. PPI has gradually become a substitute for soy protein. It is generally extracted by alkali-soluble acid precipitation method [3]. Its isoelectric point is about 4.3. The solubility of PPI is the worst when the pH is near the isoelectric point and is better under alkaline conditions. The composition of PPI includes globulin, albumin, gliadin and gluten [5], of which globulin accounts for the largest proportion accounting for about 65%-80%. Globulin is composed of legumin (11 S) and vicilin (7 S) [6]. In addition, PPI is more sensitive near the isoelectric point and prone to aggregation [7]. Studies have shown that PPI can be modified by physical, chemical and enzymatic modification methods to improve various functional properties. Sun et al. (2015) found that ultrafine pulverization and micronization of whey protein isolate improved its hydrophobicity and enabled it to have excellent emulsion stability [8].”
Point 3: In Material and methods, in pars 2.4 and 2.5 there is no need to duplicate the sentences “CS-PPI nanoparticles with ratio… Then CS-PPI nanoparticles with three…”. It is possible to refer to the statement from part 2.4 without repetition in part 2.5.
Response 3: Thank you for your comments. To avoid repetition, we have revised “CS-PPI nanoparticles solutions with three volume ratios and PPI solution were diluted with deionized water to a pea protein isolate content of 0.5% (w/v).” in part 2.5 into “CS-PPI nanoparticle solutions, PPI solution and CS solution were prepared as described in 2.4.”. Please see lines 172-173 in the revised manuscript.
Point 4: Line 131: is it correct “PPI solution” or it should be “PPI solution”?
Response 4: Thanks for pointing out this error. We have checked all the manuscript and revised the incorrect phrase “PPI solutions” into “PPI solution” in lines 164 in the revised manuscript.
Point 5: Abbreviation PDI in not introduced in Material and methods.
Response 5: Many thanks for your comment. PDI is the abbreviation of polydispersity Index. “Particle size, polydispersity Index and zeta potential determination” was revised in line 156 in the revised manuscript. Then we introduced PDI in line 157, which was shown as “The particle size, polydispersity index (PDI) and zeta potential of CS-PPI nanoparticles of all ratios, CS and PPI were measured using a high sensitive nanoparticle size analyzer (NanoBrook 90Plus, Brookhaven, USA).” Finally, the title of 3.1 was revised into “Particle size, PDI and zeta potential determination” in line 241.
Point 6: In the list of references, several corrections is needed – some journals were written in abbreviation; in some cases, journal LWT was written as Lwt and somewhere else Lwt-Food Science and Technology.
Response 6: Thank you for your comments. Referring to the published papers in journal Molecules, we have revised the journal Lwt into LWT Food Sci. Technol in lines 596, 624, 631, 633, 691 in the revised manuscript, which were marked up using the “Track Changes” function. The modified parts are as follows.
- Guo, Q.; Su, J.; Yuan, F.; Mao, L.; Gao, Y. Preparation, characterization and stability of pea protein isolate and propylene glycol alginate soluble complexes. LWT Food Sci. Technol 2019, 101, 476-482, doi:10.1016/j.lwt.2018.11.057.
- Xu, W.; Lv, K.; Mu, W.; Zhou, S.; Yang, Y. Encapsulation of α-tocopherol in whey protein isolate/chitosan particles using oil-in-water emulsion with optimal stability and bioaccessibility. LWT Food Sci. Technol 2021, 148, 111724, doi:10.1016/j.lwt.2021.111724.
- Jiang, S.S.; Hussain, M.A.; Cheng, J.J.; Jiang, Z.M.; Geng, H.; Sun, Y.; Sun, C.B.; Hou, J.C. Effect of heat treatment on physicochemical and emulsifying properties of polymerized whey protein concentrate and polymerized whey protein isolate. LWT Food Sci. Technol 2018, 98, 134-140, doi:10.1016/j.lwt.2018.08.028.
- Hadidi, M.; Motamedzadegan, A.; Jelyani, A.Z.; Khashadeh, S. Nanoencapsulation of hyssop essential oil in chitosan-pea protein isolate nano-complex. LWT Food Sci. Technol 2021, 144, 111254, doi:10.1016/j.lwt.2021.111254.
- Fan, Y.T.; Peng, G.F.; Pang, X.; Wen, Z.; Yi, J. Physicochemical, emulsifying, and interfacial properties of different whey protein aggregates obtained by thermal treatment. LWT Food Sci. Technol 2021, 149, doi:10.1016/j.lwt.2021.111904.

Reviewer 4 Report
A characterization of Pea protein isolate (PPI) and chitosan (CS) nanoparticles is performed.
The authors make a good introduction, offering the reader a good contextualization of the subject, justifying the importance of the study and offering a clear idea of the document.
The materials and methods are well described, ensuring repeatability of the study in other laboratories and scientific developments based on the information presented in this study.
The experimental quality of the data allows for a conclusive analysis of the results, in addition, the analysis is theoretically supported, and allows understanding the different phenomena related to the results obtained.
Finally, the conclusions are supported and the bibliography is adequate.
It is recommended to accept the manuscript after minor corrections.
1. Describe the information reported by the superscripts (letters) more explicitly (a=xxx, b=xxx).
Author Response
We appreciate the selfless contributions to this manuscript from you. It is because of your insightful and helpful comments and suggestions, we gained the confidence to improve our work better. The manuscript molecules-1924766 entitled “Fabrication and Characterization of Chitosan-Pea Protein Isolate Nanoparticles” has been carefully revised with the revisions marked up using the “Track Changes” function. The point-by-point answers to these comments and suggestions were listed as below.
Point 1: Describe the information reported by the superscripts (letters) more explicitly (a=xxx, b=xxx).
Response 1: Thank you for your comments. We have added the corresponding description in the caption of Fig. 4, which were shown as follow: “Different superscript letters in the figure mean significant differences (p<0.05).” in lines 513-514.

Round 2
Reviewer 1 Report
I thanks the authors for addressing my concerns. Although there are weaknesses still, for example the validity or otherwise of the electron microscopy images, I feel that the paper is now suitable for publication. It represents an interesting contribution to the study of chitosan-based composite materials.
Reviewer 2 Report
The authors have tried their best to revise the paper, and the new version was improved a lot. However, in my opinion, the design of this experiment must be improved. Before the preparation of the PPI/CS complex, the PPI solution was centrifuged at 6000 rpm for 15 min to remove any insoluble substance. As I have demonstrated, commercial PPI always shows low solubility at neutral pH. Therefore, it's reasonable to assume that large quantities of PPI proteins were discarded during the centrifugation. It was found that after centrifuging, the soluble PPI had higher stability than CS-PPI at natural pH due to the PI of the complex moving to around pH7 -8( Fig.1 ). Additionally, PPI also showed better EAI compared with CS-PPI (Fig.4A). Although the author demonstrated that the ESI of the CS-PPI complex was higher than PPI (FIg.4B), and a significant difference can be found, but in fact, less than 10% of ESI was increased by the formation of the complex. For me, it is difficult to find the theoretical and applied value of this research.